# A fast machine-learning-guided primer design pipeline for selective whole genome amplification

Jane A. Dwivedi-Yu[1,2☯], Zachary J. Oppler[3☯], Matthew W. Mitchell[3,4], Yun S. Song[1,5]*, Dustin Brisson[3]*

**1** Computer Science Division, University of California, Berkeley, Berkeley, California, United States of America, **2** Facebook AI Research, 1 Rathbone Square, London, England, **3** Department of Biology, University of Pennsylvania, Philadelphia, Pennsylvania, United States of America, **4** Coriell Institute for Medical Research, Camden, New Jersey, United States of America, **5** Department of Statistics, University of California, Berkeley, Berkeley, California, United States of America

☯ These authors contributed equally to this work.
* yss@berkeley.edu (YSS); dbrisson@upenn.edu (DB)

**Data Availability Statement:** The swga2.0 program and all dependencies can be downloaded into a conda environment from https://anaconda.org/janedwivedi/swga2. The source repository, tutorials, and documentation can be found at

## Abstract

Addressing many of the major outstanding questions in the fields of microbial evolution and pathogenesis will require analyses of populations of microbial genomes. Although population genomic studies provide the analytical resolution to investigate evolutionary and mechanistic processes at fine spatial and temporal scales—precisely the scales at which these processes occur—microbial population genomic research is currently hindered by the practicalities of obtaining sufficient quantities of the relatively pure microbial genomic DNA necessary for next-generation sequencing. Here we present `swga2.0`, an optimized and parallelized pipeline to design selective whole genome amplification (SWGA) primer sets. Unlike previous methods, `swga2.0` incorporates active and machine learning methods to evaluate the amplification efficacy of individual primers and primer sets. Additionally, `swga2.0` optimizes primer set search and evaluation strategies, including parallelization at each stage of the pipeline, to dramatically decrease program runtime. Here we describe the `swga2.0` pipeline, including the empirical data used to identify primer and primer set characteristics, that improve amplification performance. Additionally, we evaluate the novel `swga2.0` pipeline by designing primer sets that successfully amplify *Prevotella melaninogenica*, an important component of the lung microbiome in cystic fibrosis patients, from samples dominated by human DNA.

## Author summary

Population genomics enables the inference of evolutionary and ecological processes that are critical to understanding and eventually controlling many infectious diseases. The promise of microbial population genomics is tempered, however, by difficulties in isolating and preparing pathogens for next-generation sequencing. Here we present `swga2.0`, an optimized pipeline that designs the selective whole genome amplification (SWGA)

https://github.com/songlab-cal/swga2. All amplification and sequencing data presented can be found in the supplemental materials or in the Dryad data repository https://doi.org/10.5061/dryad.3n5tb2rm2.

**Funding:** This research is supported in part by the National Institutes of Health (R21-AI137433 (DB, YSS), R35-GM134922 (YSS), and R01- AI142572 (DB)), and the Burroughs Wellcome Fund (1012376 (DB)). The funding agencies had no role in study design, data collection and analysis, decision to publish, or preparation of the manuscript.

**Competing interests:** The authors have declared that no competing interests exist.

primer sets needed to amplify and sequence microbial genomic DNA from complex biological specimens.

## Introduction

The rapidly expanding field of population genomics is transforming our understanding of the evolutionary forces shaping genomic diversity within and among species [1]. In microbial systems in particular, population genomic studies are increasingly feasible due to the minimal cost of sequencing small genomes [2–5]. These studies can identify the origins of adaptive traits, map range expansions and migration patterns, and clarify epidemiological processes. A principal obstacle to sequencing specific microbial genomes from natural samples is isolating the target microbial DNA from the DNA of contaminating organisms [6]. Although laboratory culture is the standard practice, the overwhelming majority of microbes cannot be cultured, and direct sequencing is problematic as the microbial genome constitutes only a minuscule fraction of the total DNA [7–9]. Thus, a primary hindrance to collecting populations of microbial genomes is the lack of innovative, cost-effective, and practical methods to collect sufficient amounts of target microbial genomic DNA with limited contaminating DNA.

Several technologies have been developed and utilized to overcome this obstacle including genome capture, single-cell sequencing, and selective whole genome amplification (SWGA) [10–12]. Of these, SWGA is the most inexpensive, flexible, and shareable culture-free technology [13]. SWGA takes advantage of the inherent differences in the frequencies of sequence motifs ($k$-mers) among species in order to build primer sets that bind often in the target genome but rarely in the contaminating genomes. These selective primer sets are used to selectively amplify the target microbial genomes using Φ29 multi-displacement amplification technology [12, 14]. The Φ29 DNA polymerase amplifies DNA from primers with high processivity (up to 70-kbp fragments) and is 100 times less error-prone than Taq, making it the standard for genome amplification prior to sequencing [14–16]. By coupling Φ29 amplification with selective priming, researchers can selectively amplify a target microbial genome, thus separating the metaphorical baby (target microbial genomes) from the bathwater (off-target DNA from vectors, hosts, or other microbes). SWGA is a powerful and cost-effective tool for researchers looking to generate genomic data for microbial systems. Effective SWGA protocols have resulted in next-generation sequencing (NGS)-ready samples that are enriched for specific target microbial genomes and have been used to address biologically important questions in several microorganisms, including *Mycobacterium tuberculosis*, *Wolbachia spp*, *Plasmodium spp*, *Neisseria meningitidis*, *Coxiella burnetii*, *Wuchereria bancrofti*, and *Treponema pallidum* [12, 17–30].

The most recent SWGA development pipeline (`swga1.0`) improved on the concept and existing tools available for SWGA primer selection [17]. Whereas the first SWGA tool used only differential binding ratios of $k$-mers and melting temperature to build primer sets [12], `swga1.0` incorporated a larger *a priori* set of optimality criteria when selecting both individual primers (*i.e.*, primer binding frequency, improved melting temperature, evenness) and potential primer sets (*i.e.*, evenness, primer binding site density on the target genome) [17]. Evaluation of the amplification and sequencing data from this study revealed that primer sets that prioritized binding site density on the target genome, along with binding site evenness as a secondary factor, yielded the most consistent amplification success.

The `swga1.0` pipeline substantially improved the available SWGA development tools and identified several potential enhancements for future studies [17]. First, `swga1.0` uses only

marginally-effective optimality criteria to evaluate individual primers and primer sets due to a lack of empirical data of the characteristics that result in effective SWGA. While primer binding site density and evenness appear broadly important, the majority of primer sets chosen using these criteria resulted in limited amplification success. Thus, additional primer characteristics correlated with efficient selective amplification were not included in the primer selection process. Second, `swga1.0` uses a computationally-expensive algorithm to search for primer sets. `swga1.0` evaluated 1–5 million primer sets, which is only a very limited proportion of all potential primer sets, yet still required more time than is available for research projects. Evaluating primer sets could be vastly improved by an informed objective function and by pruning unpromising search paths.

The computational time and experimental cost and effort needed to develop an effective protocol to amplify a target microbial genome has hindered the broad adoption of SWGA for microbial population genomic studies. Here we present the next generation pipeline for SWGA protocol development, `swga2.0`, that improves the state-of-the-art methods in three areas. First, active learning and machine learning are incorporated into the pipeline to predict the effectiveness of primers and primer sets. Second, novel features including thermodynamically-principled binding affinities are included in primer and primer set evaluation models. Lastly, the computational efficiency of the primer set search algorithm is improved by multiprocessing and caching computationally expensive information. `swga2.0` is a fast SWGA optimization software that allows researchers to rapidly identify primer sets that are likely to amplify a specific microbial genome from a complex, heterogeneous sample. We test the novel pipeline by designing and evaluating primer sets to selectively amplify *Prevotella melaninogenica*, an important pathogen in cystic fibrosis patients.

## Methods

### `swga2.0` pipeline

The `swga2.0` pipeline incorporates metrics on the efficacy of individual primers and a computationally efficient, multiprocessing algorithm to identify sets of primers to selectively amplify a target microbial genome from samples dominated by background DNA (https://anaconda.org/janedwivedi/swga2). The `swga2.0` pipeline consists of four major stages, illustrated in Fig 1: (1) cataloging DNA sequence motifs in the target genome and identifying the locations of each motif in both the target genome and background DNA, (2) removing DNA sequence motifs that are either too rare or too unevenly distributed in the target genome, are too common in the background DNA, or have calculated melting temperatures that are outside the acceptable range, (3) predicting the amplification efficacy of the remaining primers (see **Amplification efficacy** section below), and (4) searching and evaluating aggregations of primers as candidate primer sets. A summary of similarities and differences between the `swga2.0` pipeline and prior published methods is presented in Table 1.

**Stage 1 ($k$-mer preprocessing).**   `swga2.0` first identifies all 6 bp to 12 bp $k$-mers in the target genome that serve as candidate SWGA primers. The number and location of each $k$-mer in the target genome and background DNA are computed using `jellyfish` [31]—a fast, parallel $k$-mer counter—and stored in `h5py` files. This stage of the pipeline is parallelized and does not need to be re-run when modifying parameters or generating new primer sets (Stage 4).

**Stage 2 (Candidate primer filtering).**   The $k$-mer motifs in the `h5py` files are sorted according to their binding frequency in the target genome and background DNA, the evenness of their distribution in the target genome, their calculated melting temperatures, GC content, homodimerization probability, and number of single and di-nucleotide repeats. These motif

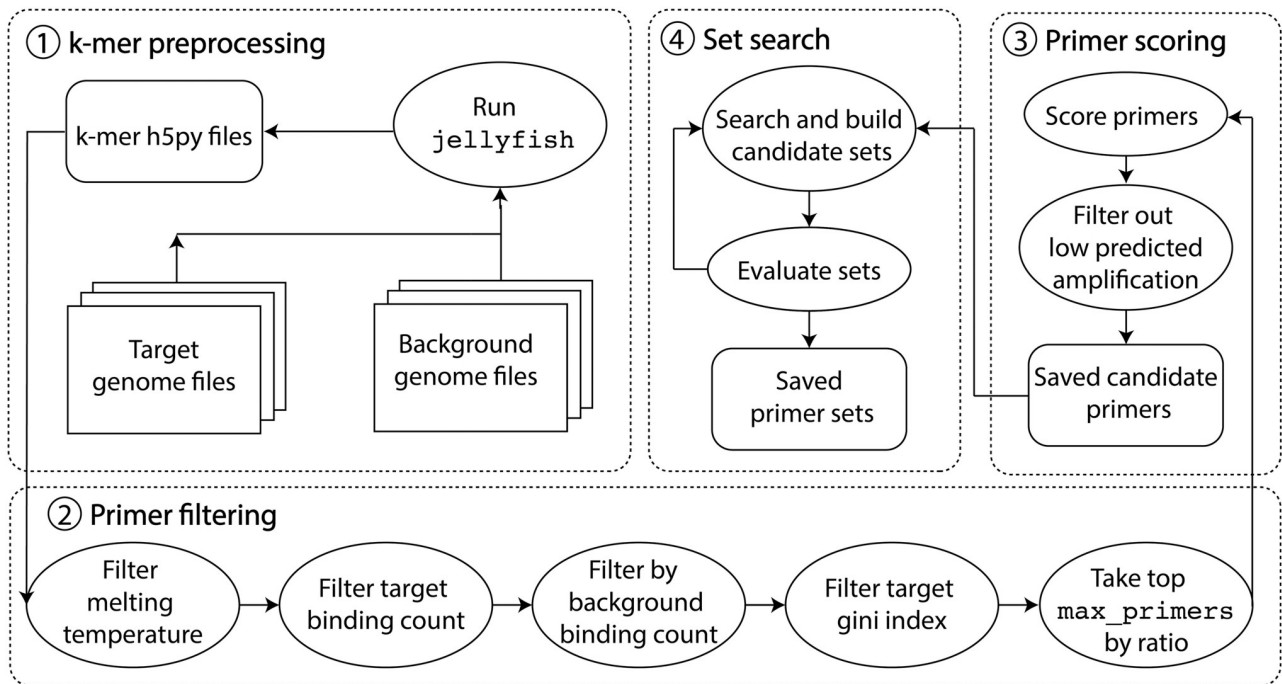

**Fig 1. Overview of the `swga2.0` pipeline.** The process is broken into four stages: 1) preprocessing of locations in the target and off-target genomes, 2) filtering motifs in the target genome based on individual primer properties and frequencies in the genomes, 3) scoring the remaining primers for amplification efficacy using a machine learning model, and 4) searching and evaluating aggregations of primers as candidate primer sets.

characteristics were used previously to identify candidate SWGA primers [17]. Briefly, binding frequency is the number of exact matches of a primer in the genome normalized by the total genome length; motifs that occur too rarely in the target genome ($<$ `min_fg_freq`) or too frequently in the background DNA ($>$ `max_bg_freq`) are removed. Primers that bind unevenly across the target genome ($>$ `max_gini`), calculated using the Gini index of the distances between motifs are removed. The melting temperature of potential primers [32] must be within `min_tm` (default 15°C) and `max_tm` (default 45°C) as established in [12]. Motifs with GC content less than `min_GC` or greater than `max_GC` (default 0.375 and 0.625, respectively), as well as motifs with three or more G/C in the last five base pairs of the 3′-end, are eliminated. Candidate primers that could self-dimerize, estimated as subsequences that have a reverse complement ($\geq$ `default_max_self_dimer_bp`, default = 4), are eliminated. Finally, motifs with runs of single and di-nucleotide repeats are eliminated. The ratio of the binding frequency in the target genome to that in the background DNA is computed for the candidate primers that remain after the filtering steps above. The primers are sorted by the target-to-background ratio and the primers with the largest ratios are retained for downstream evaluation (`max_primers`; default 500 primers). Multi-processed tasks use the user-specified number of CPUs (default = all).

**Stage 3 (Primer efficacy filter).** Decreasing the number of candidate primers reduces the computational effort necessary to search and evaluate primer sets in Stage 4. Thus, the motifs retained from Stage 2 are individually evaluated for their potential to bind and amplify the target genome, using the random forest regressor model trained on the experimental data described in the **Amplification efficacy** section below. Briefly, this non-linear regression model predicts amplification efficacy from experimentally-identified primer properties

**Table 1. Differences between `swga1.0` and `swga2.0`.**

| | Feature | `swga1.0` | `swga2.0` |
|---|---|---|---|
| All stages | Time to complete (minutes)* | 1176 | 93 |
| | Multiprocessing | | ✓ |
| | Incorporates forward strand | ✓ | ✓ |
| | Incorporates reverse strand | | ✓ |
| Stage 1 (*k*-mer preprocessing) | Time to complete (minutes)* | 100 | 20 |
| | Process *k*-mers that exact-match the target | ✓ | ✓ |
| | Process all *k*-mers in the target | | ✓ |
| Stage 2 (Candidate primer filtering) | Time to complete (minutes)* | 0.1 | 37.5 |
| | Filter by melting temp | ✓ | ✓ |
| | Filter by self-dimerization | ✓ | ✓ |
| | Filter by target binding count | ✓ | ✓ |
| | Filter by background binding count | ✓ | ✓ |
| | Filter by target Gini index | ✓ | ✓ |
| | Filter by target to background freq. binding ratio | ✓ | ✓ |
| Stage 3 (Primer efficacy filtering) | Time to complete (minutes)* | NA | 5 |
| | Filter by predicted amplification using a trained model | | ✓ |
| Stage 4 (Primer set search and eval.) | Time to complete (minutes)* | 1076 | 31 |
| | Filters based on heterodimer formation | ✓ | ✓ |
| | Score sets using heuristic function | ✓ | |
| | Score sets using fitted regression model | | ✓ |
| | Uses min distance between binding sites in the background | ✓ | |
| | Uses max distance between binding sites on the target | ✓ | |
| | Uses min ratio (target:off) of mean binding site distances | | ✓ |
| | Uses max ratio (target:off) of binding site frequency | | ✓ |
| | Uses min ratio (target:off) of coverage approximation | | ✓ |
| | Uses min mean Gini index of target binding | | ✓ |
| | Uses max mean Gini index of background binding | | ✓ |
| | Branch-and-bound search based on heterodimer cliques | ✓ | ✓ |
| | Branch-and-bound based on predict set score | | ✓ |

*Primer sets designed to selectively amplify *P. melaninogenica* (*H. sapiens* background) were built on a MacBookPro14,1 (using all 4 threads: 2.5GHz Dual-Core Intel Core i7, 2 threads per core; Memory 16GB 2133MHz LPDDR3).

including thermodynamically-principled features that correlate with binding affinity. Candidate primers are selected according to the minimum predicted on-target amplification threshold parameter (`min_amp_pred`, default = 5) in this regression model.

**Stage 4 (Primer set search and evaluation).** `swga2.0` searches for and evaluates primer sets using a machine-learning guided scoring function that incorporates a breadth first, greedy approach. The number of desired primer sets (`max_sets`) are built in parallel, primer by primer, by adding primers that cause the greatest increase in evaluation scores (see S1 Appendix). Briefly, the first primer in each of the `max_sets` sets is chosen at random from the candidate primer list allowing for broad exploration of the search space (Fig 2). Primers that are not expected to dimerize with any primer already in the set are added, one at a time, to each set and the primer set is evaluated as: score = $\beta_0 + \beta_1$ `freq_ratio` + $\beta_2$ `mean_gap_ratio` + $\beta_3$ `coverage_ratio` + $\beta_4$ `on_gap_gini` + $\beta_5$ `off_gap_gini` where the score predicts the percent of target genome coverage at 1×. The features for this regression are described in Table 2 and explained further in the section **Primer set search and scoring function**. New

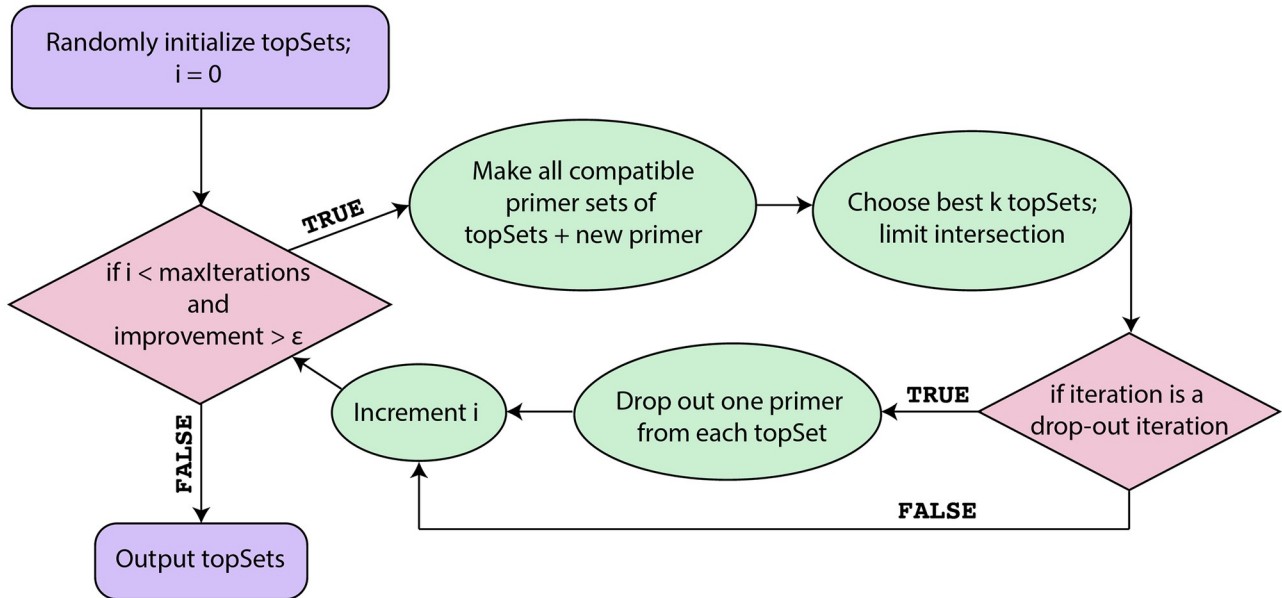

**Fig 2. Summary schematic of Stage 4 (Primer set search and evaluation) of the `swga2.0` pipeline.** Stage 4 begins with one randomly selected primer for each primer set. Each primer set is built in parallel until the improvements in evaluation score no longer exceed a user-defined parameter ($\epsilon$) or until the maximum number of iterations is reached. A drop-out iteration forces each of the highest-scoring primer sets of size $n$ to reduce to the subset of size $n - 1$ with the highest computed score.

candidate primers are added sequentially to primer sets and retained if the addition improves the computed score. This process iterates until primer sets of a desired size are generated or the maximum number of iterations is reached. The primer set search process also utilizes a drop-out step where a set of size $n$ reduces to the subset of size $n - 1$ with the highest computed score. "Dropping" the weakest primer also provides the possibility of adding a primer that would otherwise be excluded due to the risk of dimerizing with the dropped primer. The drop-out step can be re-run multiple times and includes the option of temporarily withholding frequently-used primers until after the drop-out layer.

### Amplification efficacy

The amplification efficacy of individual primers (Stage 3) is predicted using a regression model built from a series of rolling circle amplification (RCA) [14, 33] experiments in which plasmids were amplified with individual primers. SWGA typically utilizes multiple primers, each with multiple priming sites, making it challenging to isolate the impact of individual

**Table 2. Ridge regression variable descriptions and coefficient values for primer set evaluation.**

| Variable name | Variable description | Coef. | Coef. value |
|---|---|---|---|
| intercept | Intercept | $\hat{\beta}_0$ | $-3.14 \times 10^{-15}$ |
| freq_ratio | Ratio of the binding site rate in the on-target to off-target genome. | $\hat{\beta}_1$ | 0.321 |
| mean_gap_ratio | Ratio of the mean distance between binding sites of the on-target to off-target genome, aggregating across strands. | $\hat{\beta}_2$ | −0.0368 |
| coverage_ratio | Ratio of the coverage approximation of the on-target to that of the off-target. | $\hat{\beta}_3$ | −0.0318 |
| on_gap_gini | Mean gini index of on-target binding site gap sizes, averaging across strands. | $\hat{\beta}_4$ | −0.0131 |
| off_gap_gini | Mean gini index of off-target binding site gap sizes, averaging across strands. | $\hat{\beta}_5$ | 0.281 |

primers. Single-primer amplification reactions assess amplification efficacy of individual primers binding from plasmids with or without an exact-match binding site.

The model of primer amplification efficacy included the features delineated in Table 3 as parameters. Some of these attributes include properties such as the number of base pair repeats, melting temperature and G/C proportion, all of which are thought to impact the efficacy of accurate primer binding and amplification in PCR and Φ29 reactions [34]. `swga2.0` includes additional features that estimate the likelihood of the primer binding to the target using a unified thermodynamic nearest-neighbor DNA model [35, 36]. Empirical thermodynamic parameters ($\Delta G_T^\circ$) are available for most primer exact-match and single-mismatch scenarios. Empirical thermodynamic data for terminal mismatches are not publicly available [35, 36] and were not captured in our predictive model. This thermodynamic nearest-neighbor model was incorporated by computing $\Delta G_T^\circ$ values for each primer at each genome position—a smoother metric for primer binding propensity than the number of exact match binding sites. The $\Delta G_T^\circ$ values were binned within the range of −20 and 3, and the resulting histogram normalized by genome length. The normalized $\Delta G_T^\circ$ frequency values are used as features in the primer amplification efficacy regression model.

**An active learning approach.**   Active learning, a type of iterative supervised machine learning, was used to maximize information gain in three rounds of single primer amplification experimentation. The previously published SWGA Perl script [12] was used to generate a list of all primers with one exact-match binding site on one of the plasmids and no exact-match binding site on the other plasmid (pcDNA3-EGFP and pLTR-RD114A from Addgene). The first round of experimental amplification used 204 primers from this list that maximized the variability in the 22 primer attributes in Table 3, excluding the thermodynamic binding affinity features. A random forest regressor model was built using the target and off-target amplification data from the first round of experimental amplifications as it had the best performance of the tested models (linear, logistic, random forest, gradient boosting, and support vector machine). The optimal parameters according to a hyperparameter search were `n_estimators`=1500, `min_samples_split`=10, `min_samples_leaf`=4, `max_depth`=50, `bootstrap`=False.

The random forest regression model was used to predict the amplification efficacy of all primers in the original list. The 96 primers predicted to have the greatest amplification efficacy were chosen for the second round of experimental assessment. The experimental amplification data from round 2 were used to update the random forest regression model. An additional 96 primers predicted to have the greatest amplification efficacy were chosen for a third round of experimental evaluation. The final random forest regression model, built on the experimental results from three rounds of single-primer amplification experiments, is included in the `swga2.0` primer design pipeline (Stage 3).

## Primer set scoring function

Multiple analytical frameworks were explored to construct a primer set scoring function that would accurately predict amplification efficacy and evenness from an individual primer set, determined by the proportion of the target genome with at least 1× sequencing coverage. Multiple primer set scoring functions were trained on data from 46 sets of published SWGA and sequencing data from *Mycobacterium tuberculosis* and *Homo sapiens* [17]. Model features were selected from a set of variables thought to influence amplification of the target genome (Gini index, nucleotide distance between target binding sites, entropy and generalized entropy of the binding site distribution), amplification of background DNA (kurtosis, skewness, bimodality, and variance among binding sites), and combinations of target to background

**Table 3. Feature importances based on the random forest regressor model.**

| Subset Description | Feature Description | Feature Importance (%) | Subset Feature Importance (%) |
|---|---|---|---|
| G/C content features | number/proportion of G's | 11.8/2.68 | 27.9 |
| | number/proportion of C's | 2.76/6.54 | |
| | GC content | 4.08 | |
| repeat features | GG repeat number | 7.30 | 19.2 |
| | longest G repeat | 4.10 | |
| | CC repeat number | 2.37 | |
| | longest C repeat | 3.51 | |
| | TT repeat number | 0.458 | |
| | longest T repeat | 0.613 | |
| | AA repeat number | 0.333 | |
| | longest A repeat | 0.431 | |
| Binding affinity features | 3 | 1.42 | 18.1 |
| | 2.5 | 1.70 | |
| | 2 | 0.942 | |
| | 1.5 | 1.84 | |
| | 1 | 2.07 | |
| | 0.5 | 1.21 | |
| | 0 | 1.27 | |
| | −0.5 | 0.778 | |
| | −1 | 0.616 | |
| | −1.5 | 0.532 | |
| | −2 | 0.368 | |
| | −2.5 | 0.413 | |
| | −3 | 1.25 | |
| | −3.5 | 0.633 | |
| | −4 | 0.611 | |
| | −4.5 | 0.105 | |
| | −5 | 0.0120 | |
| | −5.5 | 0.0143 | |
| | −6 | 0.134 | |
| | −7 | 0.0768 | |
| | −8 | 0.259 | |
| | −9 | 0.0799 | |
| | −10 | 0.215 | |
| | −12 | 0.531 | |
| | −14 | 0.667 | |
| | −16 | 0.199 | |
| | −18 | 0.0802 | |
| molarity | molarity | 10.2 | 10.2 |
| last 5 bases near 3′-end | GC-clamp | 1.92 | 10.2 |
| | first base from 3' end | 1.08 | |
| | second base from 3' end | 2.48 | |
| | third base from 3' end | 1.32 | |
| | fourth base from 3' end | 1.41 | |
| | fifth base from 3' end | 1.96 | |
| A/T content features | number/proportion of A's | 1.16/1.25 | 6.42 |
| | number/proportion of T's | 1.03/2.98 | |

(*Continued*)

**Table 3.** (Continued)

| Subset Description | Feature Description | Feature Importance (%) | Subset Feature Importance (%) |
|---|---|---|---|
| melting temperature | melting temperature | 6.33 | 6.33 |
| sequence length | sequence length | 2.56 | 2.56 |

amplification (the frequency of binding sites, average distance between binding sites on the same strand, average distance between binding sites on the opposite strands). The best model was selected using 10-fold cross-validation error among ridge regression candidate models. Ridge regression, which uses regularization to reduce model variance, was favored in order to reduce the risk of overfitting given the limited data.

## Empirical evaluation of primer sets

Primer sets designed to amplify *Prevotella melaninogenica* from samples dominated by human DNA were empirically tested to evaluate the efficacy of `swga2.0`. Six primer sets were created using *P. melaninogenica* strain ATCC 25845 (GCF_000144405.1) as the target genome and the human genome (GRCh38.p13) as the background DNA. The six primer sets were evaluated in duplicate on purified *P. melaninogenica* DNA (strain ATCC 25845), diluted to 1% in purified human genomic DNA (Promega, female, catalog No. G1521). Briefly, the 1:99 target:background sample was digested with FspEI (New England Biolabs) according to the manufacturers protocol (incubation at 37˚C for 90 minutes, 20 minute heat inactivation at 80˚C). Although digestion has reduced mitochondrial DNA amplification or increased target amplification in some prior studies [12, 20, 37], experiments amplifying non-digested DNA using two of the *Prevotella* primer sets resulted in comparable amplification success with the digested samples. The digested sample was purified using AmpureXP beads (Beckman Coulter) prior to performing selective whole-genome amplification as previously described [18] with slight modifications. Reactions were performed in a volume of 50 uL using 50 ng of digested DNA, SWGA primers (total concentration of all primers together = 3.5mM), 1× Φ29 buffer (New England Biolabs), 1 mM dNTPs, and 30 units Φ29 polymerase (New England Biolabs). Amplification conditions included a ramp-down from 35 to 31˚C (5 min at 35˚C, 10 min at 34˚C, 15 min at 33˚C, 20 min at 32˚C, 25 min at 31˚C), followed by a 16h amplification step at 30˚C. The polymerase was then denatured for 15 min at 65˚C. Amplified samples were purified using AmpureXP beads, prepared for Illumina sequencing [38], and sequenced on an Illumina MiSeq (150 bp, paired end). The unamplified sample was also sequenced to assess changes in sequencing coverage due to SWGA. Illumina-specific adapter and primer sequences were removed from the reads using `Cutadapt` (Martin, 2011) and reads aligned to the target genome using `BWA mem` (v0.7.1). Analysis of sequence coverage of the target genome was performed using `samtools` (v1.9) [39].

## Results

### Primer amplification efficacy

Individual primers were evaluated in amplification experiments using plasmids that either did or did not have exact-match binding sites. The active learning approach, in which data were collected to train a random forest regression model that was then used to choose additional primers, maximized the information gained from each of the three rounds of amplification experiments. The goal of the first round of experiments was to maximize the exploration of the

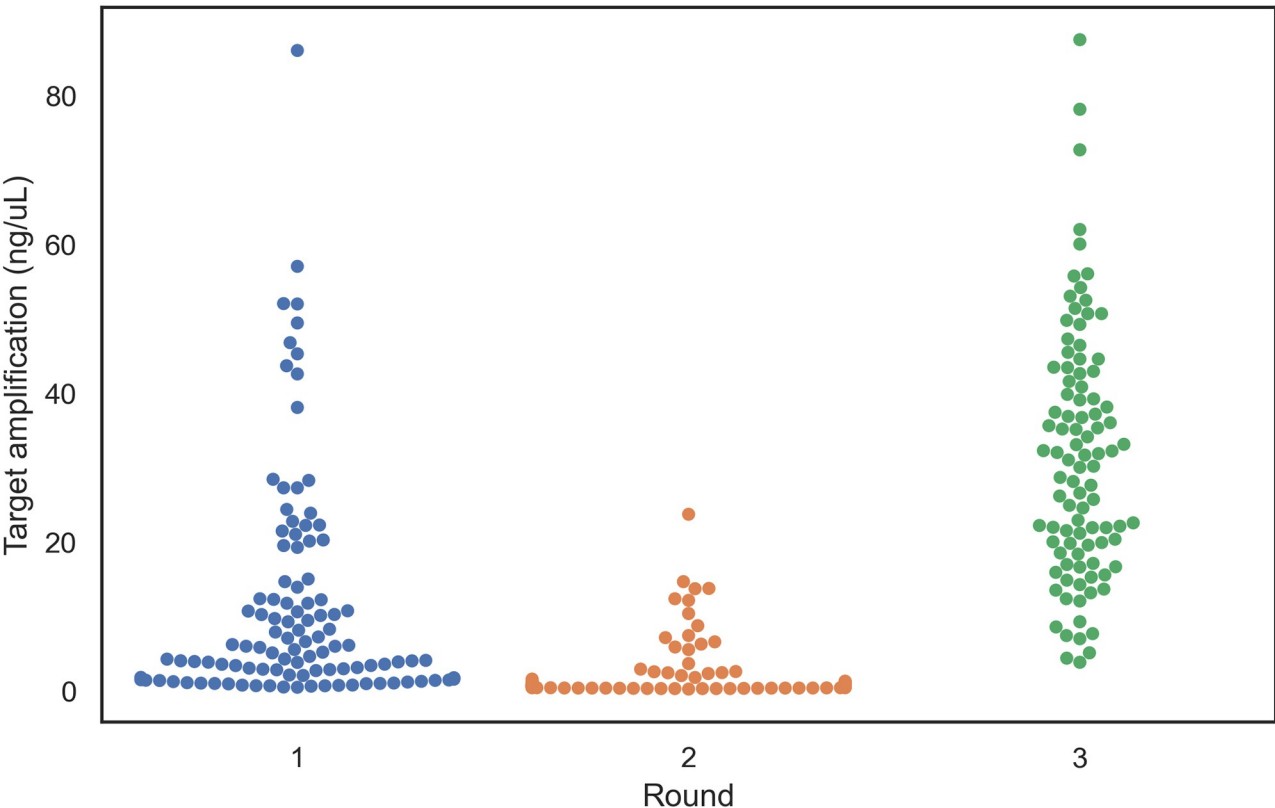

**Fig 3. The accuracy of amplification efficacy predictions increased after three iterations of the active learning approach designed to identify primer characteristics associated with effective priming.** Amplification of the target plasmid was weak in the majority of the randomly selected primers experimentally investigated in Round 1 (blue points represent the 204 primers evaluated). Target plasmid amplification was equally poor for the primers selected for Round 2 experimentation (96 orange points) by the random forest regressor model trained on the data from Round 1. The majority of primers selected for Round 3 experimentation (96 green points) by the updated random forest regressor model trained on the data from Rounds 1 and 2 resulted in moderate and high amplification of the target plasmid. Points are adjusted along the *x*-axis so that they do not overlap.

feature space across 22 primer characteristics (Table 3, excluding binding affinity features). Only a few of the 204 tested primers resulted in strong amplification of the target plasmid and weak amplification of the off-target plasmid (Fig 3; S1 Table).

The data collected from the first round of amplification experiments were used to build a random forest regression model and predict the amplification potential of an additional 96 primers (S1 Table). Although a greater proportion of the primers utilized in the second experimental round resulted in strong amplification of the target plasmid, the majority still performed poorly (Fig 3), likely due to training the initial model with few high-performing primers from the initial round which limited extrapolation to the high-amplification regime. Training the random forest regression model with data from rounds 1 and 2 resulted in a model that predicted primarily high-amplification primers.

The high variance in the amplification efficacy of primers selected by the final random forest model iteration makes accurate prediction of high-amplification primers difficult. Fortunately, however, this model accurately predicts poor amplification (amplification scores < 10; Fig 4). As the utility of this model is to exclude low-amplification primers from SWGA primer sets, the `swga2.0` pipeline uses the random forest regressor to filter low-amplification primers in Stage 3 (Primer efficacy filter). The random forest regressor excludes a high proportion of low-performing primers and rarely excludes higher-performing primers as determined by

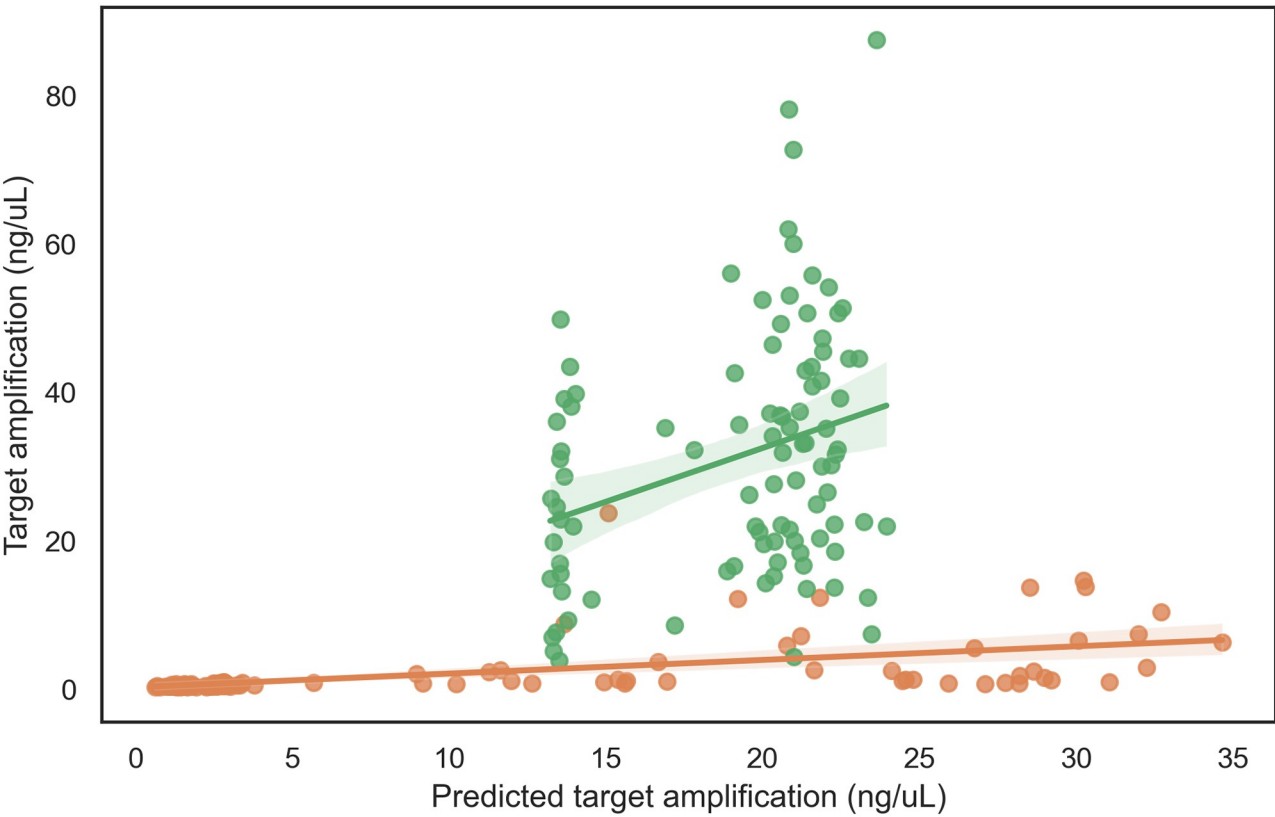

**Fig 4. The final random forest regression model reliably predicts the amplification efficacy of individual primers.** The amplification predicted for Round 2 amplification was accurate only for low-performing primers (orange). Updating the model with Round 2 amplification data resulted in a model that predicted highly effective primers (green). The amplification efficacy of the primers selected for Round 3 were highly variable despite similar predictions. Nevertheless, the final random forest regression model did not select any poor performing primers. The model was not used to predict the primers used in Round 1.

testing the model predictions on out-of-sample data (Table 4). The minimum predicted amplification parameter (`min_amp_pred`, default threshold value = 5) can be modified to exclude a greater proportion of poor-performing primers. Excluding greater numbers of poor-performing primers will reduce the computational complexity of finding primer sets and reduce the probability of experimentally evaluating poor primer sets.

**Feature importance in the random forest regression model.** Feature importance, computed from the variance reduction at each split in each tree, identifies the impact of each variable on primer performance predictions (Table 3). Features involving GC content are the most important subset of features (27.9%) followed by binding affinities calculated from the

**Table 4. Poor performing primers are filtered out in Stage 3 of `swga2.0`.** Higher threshold values in the random forest regression model filter greater proportions of lower-amplification primers but few moderate or efficient primers.

| Threshold parameter | Total primers filtered | High-amplification primers filtered |
|---|---|---|
| 2 | 6.5% | 0.04% |
| 5 | 26.5% | 1.6% |
| 10 | 47.1% | 3.8% |
| 15 | 58.9% | 4.4% |
| 20 | 66.6% | 5.4% |

thermodynamic binding model (18.1%). The features involving GC-content and the thermodynamic binding affinity features are in agreement, as expected, as these sets of features are correlated. For example, primers with greater GC tend to have more negative $\Delta G_T^\circ$ values calculated in the thermodynamic model [35].

## Primer set search and scoring function

The optimal primer set scoring function selected uses five variables (`freq_ratio`, `mean_gap_ratio`, `coverage_ratio`, `on_gap_gini` and `off_gap_gini`) within a ridge regression framework (Table 2). The variables `freq_ratio` and `mean_gap_ratio` are summary statistics previously found to correlate with SWGA success [17]. The `freq_ratio` variable is a simple measure of binding site frequency between the on-target and off-target genomes. The `mean_gap_ratio` variable measures the average distance in base pairs between primer positions on both the forward and reverse strands where the computation is indifferent to the strand on which a binding position lies. `coverage_ratio` is computed by first identifying the number of binding sites on the opposite strand that are within 70 kbp of an exact primer binding site (70 kbp is the maximum length of the synthesized DNA by Φ29 [40]). Exponential amplification using Φ29 requires priming positions in the opposite direction within 70 kbp of each other. The number of binding sites within 70 kbp of each binding site on the opposite strand is then normalized by the total genome length as a proxy for primer 'coverage'. The `coverage_ratio` is the ratio of the coverage in the target genome to the coverage in the background DNA, which is minimized in the ridge regression model (negative regression coefficient value). Primer evenness is represented in the model for each strand of the target and off-target genomes by `on_gap_gini` and `off_gap_gini`, respectively, and is computed using the Gini index of distances between binding sites.

**Computational costs are reduced more than ten-fold.** The computational time needed to build primer sets using `swga2.0` is significantly lower than for the original `swga1.0` program (Table 1). Both programs use the same base framework to create files to store the catalog of 6–12-mers found in the target and background genomes. In `swga2.0`, however, these files are stored efficiently for future use and need not be re-created if additional primer sets are desired. Using a 2013 MacBook Pro (2 GHz Intel Core i7 with 16 GB of memory; `max_sets` = 5; `max_primers` = 200), the `swga2.0` pipeline completes its primer set design for *P. melaninogenica* and *H. sapiens* in 93 minutes, a process that requires more than ten times as long using `swga1.0` (Table 1). The increase in computational efficiency is the result of multi-processing in many aspects of the pipeline and utilization of an efficient rather than exhaustive search algorithm. Additionally, file formats like `h5py` allow for O(1) read access to primer binding positions and data structures that cache primer set scores which eliminates unnecessary recalculations for future iterations. Lastly, `swga2.0` uses an efficient expression to calculate the exact Gini score and takes advantage of efficiencies in array computations in the Python library `numpy`.

## Evaluation of primer sets to selectively amplify *Prevotella melaninogenica*

*Prevotella melaninogenica* is an important pathogen in cystic fibrosis patients that is difficult to culture from human-derived samples [41, 42]. *P. melaninogenica* was chosen to evaluate the improvements in the `swga2.0` primer design pipeline as it was expected to be more challenging to selectively amplify for several reasons. For example, both the human genome and *P. melaninogenica* have a GC-content of 41% while the GC-content in the *M. tuberculosis* genome is 65.6% [43, 44]. Due to the similarities between *P. melaninogenica* and the human genome, only 14 candidate primers passed the most restrictive filters in Stage 2 (`min_fg_freq` = 1/33,333;

**Table 5. _P. melaninogenica_ is a more difficult genome to design primers for than _M. tuberculosis_.** Searching the two genomes with the same parameters during Stages 2 and 3 produces much larger lists of candidate primers for _M. tuberculosis_ than for _P. melaninogenica_. Values are the number of candidate primers remaining after Stage 3 (parenthetical numbers are the candidate primers remaining after Stage 2).

| Mean Target Frequency | Mean Background Frequency | _Prevotella melaninogenica_ | _Mycobacterium tuberculosis_ |
|---|---|---|---|
| < 1/33.3 kbp | > 1/500.0 kbp | 14 (84) | 114 (676) |
| < 1/33.3 kbp | > 1/333.3 kbp | 19 (143) | 162 (884) |
| < 1/33.3 kbp | > 1/300.0 kbp | 22 (174) | 169 (939) |
| < 1/40.0 kbp | > 1/500.0 kbp | 48 (217) | 168 (920) |
| < 1/40.0 kbp | > 1/333.3 kbp | 93 (396) | 250 (1238) |
| < 1/40.0 kbp | > 1/300.0 kbp | 103 (448) | 260 (1308) |

max_bg_freq = 1/500,000). By contrast, 114 primers passed the same set of filters for _M. tuberculosis_ (Table 5). The number and quality of primers retained in Stages 1–3 of the swga2.0 pipeline is critical to building effective primer sets to selectively amplify a target genome in Stage 4. We built three primer sets for _P. melaninogenica_ from the 19 primers retained using a less restrictive background DNA filter (Primer sets 1–3; min_fg_freq = 1/33,333 bp; max_bg_freq = 1/333,333 bp) and three primer sets from the 48 primers retained using a less restrictive target genome filter (Primer sets 4–6; min_fg_freq = 1/40,000 bp; max_bg_freq = 1/500,000 bp; Table 5). The primer sets and their associated statistics are presented in S2 Table.

Although the sequence motif similarities between _P. melaninogenica_ and the human genome made primer set design more difficult, three of the six primer sets built by swga2.0 were highly effective at selectively amplifying _P. melaninogenica_ from a sample dominated by human DNA (99%) (S3 Table; https://doi.org/10.5061/dryad.3n5tb2rm2). By comparison, 40% of the primer sets designed to amplify _M. tuberculosis_ using the prior swga1.0 pipeline performed much better than the unamplified controls [17]. High-throughput sequencing of the amplification products from the three effective _P. melaninogenica_ primer sets reached 1× coverage across 25–64% the target genome with 50 Mbp sequencing effort (Fig 5). The most effective primer set designed by swga1.0 in a prior study reached 1× coverage across only 27% of the _M. tuberculosis_ with 50 Mbp sequencing effort [17]. Similarly, deeper sequencing of the amplification products of the effective primer sets reached 10× coverage at 33–82% of the the _P. melaninogenica_ genome after 700 Mbp of sequencing effort, compared with just 0.2% for the unamplified control (Fig 6). The most effective _M. tuberculosis_ primers reached 10× coverage at less than 30% of the target genome at similar sequencing effort [17].

## Discussion

Analyses of populations of microbial genomes has the power to address many major outstanding questions in evolution and pathogenesis. Obtaining populations of genome sequence data has been aided by practical and cost-effective method advancements like Selective Whole Genome Amplification (SWGA). However, developing and verifying an effective SWGA primer set to selectively amplify a target genome from a heterogeneous sample has been both computationally and experimentally challenging. Prior applications of SWGA required considerable computational investment followed by experimental assessments of numerous primer sets, of which only a few worked sufficiently well. The swga2.0 pipeline efficiently identifies primer sets, of which half are highly effective. These improvements were achieved by experimentally identifying characteristics of individual primers that result in effective amplification and limit mispriming and by using efficient data structures and search algorithms to

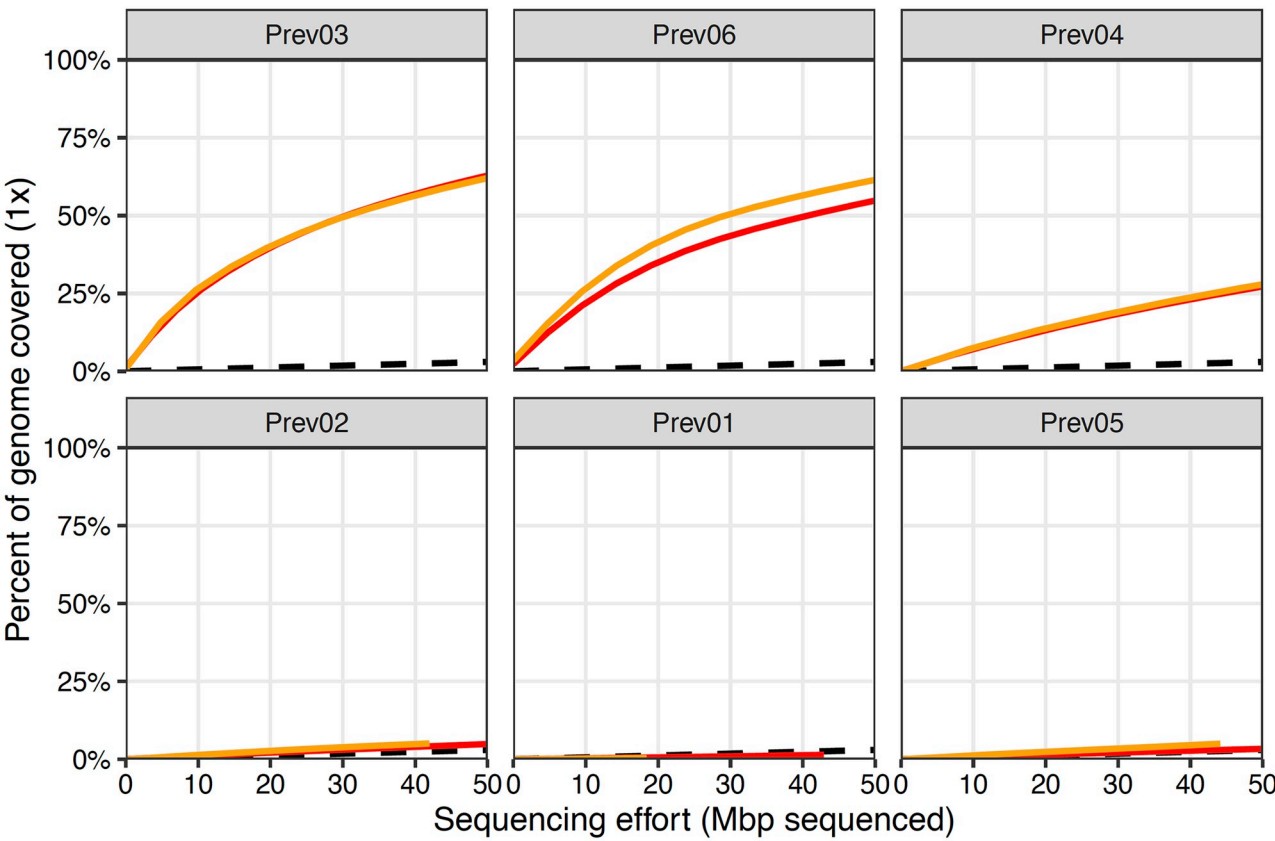

**Fig 5. Selective amplification of *P. melanogenica* using six `swga2.0`-designed primer sets (Prev01-Prev06) identified three primer sets (Prev03, Prev06, and Prev04) that can amplify the target genome but not the background DNA.** Red and yellow lines indicate the percent of the target genome covered at 1× depth from sequencing each of the two replicate amplification experiments. The black dashed lines represent sequencing coverage of the unamplified samples. While five of the six sets resulted in greater sequencing coverage of the target genome compared to unamplified controls, two were only marginally better. By contrast, sequencing coverage of the amplicons from three of the sets was substantially better than the coverage of the unamplified samples.

identify primer set characteristics that are correlated with strong and even amplification. The resulting `swga2.0` program reduces both the computational and experimental investment necessary to design and subsequently utilize protocols to effectively transform a complex sample containing mostly DNA from nontarget species to a sample dominated by the genomic DNA of the target species.

Computational identification of SWGA primer sets is a highly complex optimization problem involving scalability challenges and limited, noisy prior data. The previously published `swga1.0` program identified over five million candidate primers that needed to be filtered into a reasonable working catalog prior to designing primer sets to selectively amplify *M. tuberculosis*. The computational effort needed to assess each primer set combination requires data structures that efficiently store the information needed for evaluation without repeated searches across multiple genomes for potential binding locations. Further, identifying search and evaluation strategies to identify primer sets demands combinatorial optimization techniques that could be aided through analyses of prior SWGA data. However, learning from prior data has its own challenges due to both limited data availability and experimental noise associated with short-read sequencing data. Despite these challenges, the `swga2.0` program

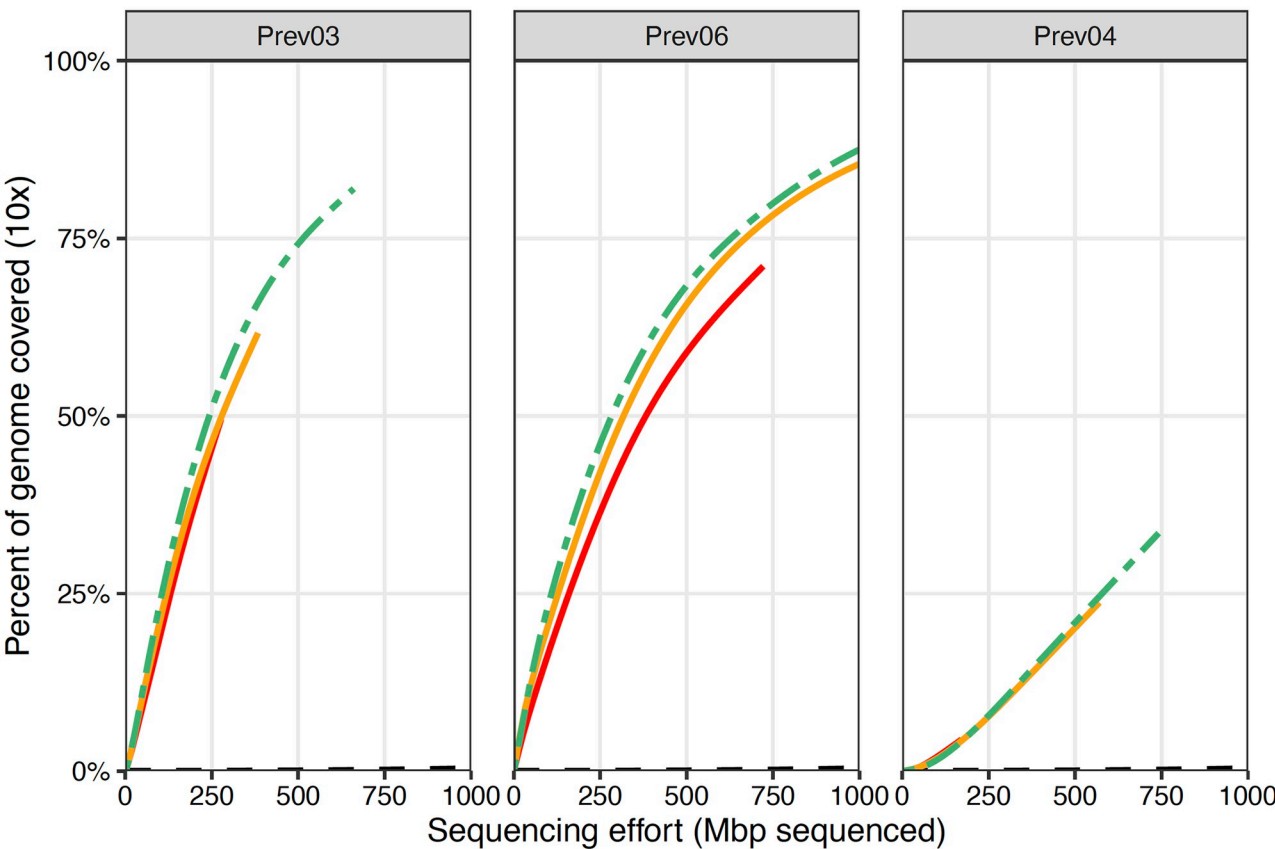

**Fig 6. Deeper sequencing of the three successful primer sets—Prev03, Prev06, and Prev04—confirms the efficient and even selective amplification of _P. melaninogenica_.** The solid colored lines indicate individual replicates and the green dashed line represents the pooled total. Each of the three primer sets yield dramatic increases in sequencing depth compared to the unamplified samples (black dashed line). Each primer set reached 10× coverage across 23–74% of the target genome, while the unamplified samples reached 10× coverage at <1% of the target genome, with 500 Mbp of sequencing effort.

provides the necessary pipeline to rapidly identify primer sets that have a high probability of selectively amplifying any target genome from any background DNA sample [45].

The swga2.0 program offers a number of improvements including dramatically reduced computation time due to the implementation of parallelization, caching, branch-and-bound searching, and efficient data storage formats (Table 1). Each of the four stages of the computational pipeline improves speed and accuracy. For example, the jellyfish program [31] used in _k_-mer preprocessing (Stage 1) is much faster than the DSK program [46] used in swga1.0; novel filters improve primer identification and prevent self-dimerization in filtering candidate primers (Stage 2); a novel machine-learning model for scoring individual primers based on amplification efficacy, which includes thermodynamically-principled binding affinities, improves primer efficacy estimation and reduces the candidate primer list (Stage 3); and the implementation of branch-and-bound techniques with randomized starting locations, drop-out techniques, and additional primer set evaluation functions learned from prior data improve both speed and accuracy of primer set searches and evaluation (Stage 4). Most importantly, optimization is no longer done by hand or via exhaustive search [12, 17].

Amplification from 6 to 12 bp primers is prone to mispriming. The random forest model presented here, built on iteratively collected experimental data, accurately identifies low-

amplification primers with very few false positive identifications. Applying this filter to the candidate primers from `swga1.0` identifies more than 85% as having low-amplification potential. Further, all of ten top-rated primer sets designed by `swga1.0` are made exclusively of primers with low-amplification potential. The random forest model is generalizable to all SWGA projects as it identifies amplification efficacy from primer sequence properties such as GC content and thermodynamically-principled binding affinities [35, 36]. Thus, we expect that this model can be used to predict effective primers when using alternative polymerases such as *EquiPhi29* or *Bst*. However, empirical validation of the primer efficacy model using alternative polymerases would be prudent. Regardless, this novel model allows researchers to remove low-amplification primers from the primer set search and evaluation (Stage 4), resulting in a considerable reduction in computational time complexity and reducing the number of experimentally-tested primer sets that perform poorly.

The primer sets designed by `swga2.0` to selectively amplify *P. melaninogenica* from a sample dominated by human DNA were highly successful. Out of the six primer sets tested, three sets amplified the target genome significantly more than the negative controls. Similar to the conclusion from prior primer set design algorithms [12, 17], the primer sets with lower mean binding distance in the target genome (Prev03—1/4.1 kbp, Prev04—1/2.0 kbp, Prev06—1/4.9 kbp) generally outperformed the other sets (Prev01—1/7.4 kbp, Prev02—1/5.0 kbp, Prev05—1/2.8 kbp). However, all of the sets have similar mean binding distances suggesting that other primer or set attributes that have not been identified likely account for much of the observed variation in amplification success (S2 Table). Nevertheless, it is expected that approximately half of all designed sets are likely to give strong and even selective amplification results such that few sets need to be experimentally evaluated by researchers. Also similar to prior results [20], protocols developed with `swga2.0` are expected to retain the ability to investigate within-host microbial diversity using SWGA enrichment [45], without introducing errors, as the SWGA biochemistry remains identical (S4 Table). These advances reduce the up-front cost of developing the SWGA primer sets necessary for population genomic studies.

The data and analyses from this and prior SWGA projects suggest the following as a general primer identification and primer set design workflow for future projects:

- Stage 2 (Candidate primer filtering): Set the `min_fg_freq` parameter as high as possible, and the `mix_bg_freq` parameter as low as possible but not below $3 \times 10^6$, while retaining approximately 100 primers.

- Stage 3 (Primer efficacy filter): Set the `min_amp_pred` to 10 or higher to eliminate low-efficiency primers. If too few candidate primers are retained ($< 20$), increase the number of primers that pass Stage 2 filtration by reducing `min_fg_freq` as opposed to reducing the `min_amp_pred` parameter value.

- Stage 4 (Primer set search and evaluation): Select 5–10 primer sets with the highest scores for experimental evaluation. However, it is not recommended to choose the top 5–10 sets as they often differ by only one or two primers. It is prudent to choose three of the top-scoring sets and several others with high scores that share few primers with the top-scoring sets. Experimentally amplify the target genome from a mixed sample ($\approx 1\%$ target DNA), barcode amplicons from each primer set separately, then pool and sequence at low depth to assess performance. Sequence amplicons from high-performing sets to ensure quality and evenness.

Best practices are likely to evolve as SWGA is used more frequently. To facilitate this, the project source repository web page contains on a tutorial on the program's operation and

more extensive documentation on each parameter and module as well as a link to a user mailing list (https://github.com/songlab-cal/swga2).

## Dryad DOI

https://doi.org/10.5061/dryad.3n5tb2rm2.

## Supporting information

**S1 Appendix. Primer set search algorithm.**
(PDF)

**S1 Table. Primer amplification data.**
(PDF)

**S2 Table. Primer set statistics and sequences.**
(PDF)

**S3 Table. Percent of reads mapping to *Prevotella* and Human (background) sequences (15 Mbp sequenced).**
(PDF)

**S4 Table. Proportion of called bases matching the *Prevotella* reference genome demonstrates that SWGA does not introduce sequencing errors.**
(PDF)

## Acknowledgments

We are grateful to Paul Planet and Prioty Sarwar for providing genomic DNA.

## Author Contributions

**Conceptualization:** Jane A. Dwivedi-Yu, Matthew W. Mitchell, Dustin Brisson.

**Data curation:** Zachary J. Oppler.

**Formal analysis:** Jane A. Dwivedi-Yu, Yun S. Song.

**Funding acquisition:** Yun S. Song, Dustin Brisson.

**Investigation:** Jane A. Dwivedi-Yu, Zachary J. Oppler, Matthew W. Mitchell, Yun S. Song, Dustin Brisson.

**Methodology:** Zachary J. Oppler, Matthew W. Mitchell, Yun S. Song.

**Project administration:** Dustin Brisson.

**Resources:** Matthew W. Mitchell.

**Software:** Jane A. Dwivedi-Yu.

**Supervision:** Yun S. Song, Dustin Brisson.

**Writing – original draft:** Jane A. Dwivedi-Yu, Zachary J. Oppler, Matthew W. Mitchell, Yun S. Song, Dustin Brisson.

**Writing – review & editing:** Jane A. Dwivedi-Yu, Zachary J. Oppler, Matthew W. Mitchell, Yun S. Song, Dustin Brisson.

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
