## [Decision Letter · Decision Letter 0]

18 Jul 2022

Dear Dr. Brisson,

Thank you very much for submitting your manuscript "A fast machine-learning-guided primer design pipeline for selective whole genome amplification" for consideration at PLOS Computational Biology.

As with all papers reviewed by the journal, your manuscript was reviewed by members of the editorial board and by several independent reviewers. In light of the reviews (below this email), we would like to invite the resubmission of a significantly-revised version that takes into account the reviewers' comments.

We cannot make any decision about publication until we have seen the revised manuscript and your response to the reviewers' comments. Your revised manuscript is also likely to be sent to reviewers for further evaluation.

Sincerely,

Manja Marz

Software Editor

PLOS Computational Biology

Manja Marz

Software Editor

PLOS Computational Biology

Reviewer's Responses to Questions

**Comments to the Authors:**

Reviewer #1: In this manuscript the authors introduce version 2 of the SWGA pipeline which optimizes primer design for selective whole genome amplification. This is a very valuable tool for enriching target DNA for sequencing in cases where samples would typically be dominated by background or host DNA. In the new version, the authors have added several speed ups to the original algorithm and used a machine learning approach to identify efficient primers features. The new additions to the pipeline are substantial and warrant a second paper to introduce these enhancements. The authors demonstrate the performance of swga2.0 by designing and testing 6 primer sets for amplifying Prevotella melaninogenica within a human background. The manuscript is well organized and clearly written in a way that is understandable to a non-computational audience and the best practices section at the end is very helpful.

The authors provide publicly available, version-controlled, and well documented code with appropriate documentation on installation and workflow. A local copy of the codebase can easily be installed using pip however it would be assessable to a wider audience if it could be installed using Conda (or similar) with all dependencies. The name of the software is different than what is reported in the paper which may be confusing. There does not appear to be any unit tests.

The sequencing and raw data was not provided with the manuscript and would be required to reproduce the results. The primer sequences were also not provided as far as I can tell.

There are not many other tools to directly compare to, so the authors mostly refer to previous results of swga1.0 but they did not do a direct comparison. Because the focus of this paper is on improvements from version 1.0, it would be helpful if the authors provide a summary table that compares the features between the old and new versions. To drive home the benefit of the newly added primer efficacy filter, it would be good to build sets using swga1.0 and evaluate how often the “best” sets contain primers that are predicted to be low efficiency and at what proportion. This would lend support the claims in the paper and would be fairly quick to run without requiring any additional sequencing.

Furthermore, given that the multiprocessing speedups are another focus of the paper, I recommend that a more rigorous comparison is performed between the two versions. A time was provided for a single run of v2.0 and the time for v1.0 was given as over two weeks. The number of processors and the memory requirements were not provided. It would be good to know how many processors were used and if there are diminishing returns. Was swga1.0 able to find a result after running for 2 weeks or did it get stuck? I believe that I have been able to design SWGA primers in less than two weeks using swga1.0, so it is unclear why it took so long. Perhaps it would be good to try a less complicated design to compare the efficiency between the two versions so that we can better evaluate the improvement of each of the speedups. It would be great to see a breakdown of the estimated time saved due to each of the improvements listed in the section titled: “Computational costs are reduced from weeks to minutes”

About half of the primer sets failed using swga2.0. There was some mention in the discussion about a relationship between the successful sets and lower mean binding distance in the target genome, but it would be nice to have a table of all the observed stats for the sets in addition to the parameter settings.

The discussion mentions that this approach would work on metagenomic samples (last paragraph). Has this been done and how would swga2.0 be able to design primer sets for such complex backgrounds? How many background genomes could reasonably be added? How would you handle all the potential plasmids in such a sample?

The last paragraph of the discussion abruptly introduces a number of new ideas that are difficult to follow. These should be expanded on to provide clarity if they are important or eliminated.

The EquiPhi29 enzyme is more thermally stable and produces higher yield than Phi29. Using this enzyme may allow longer primers with higher Tm. Could the authors provide some discussion about whether swga2.0 can design appropriate primer sets for use with this enzyme and some recommended parameter adjustments.

Mispriming in the rolling circle plasmid amplification experiment was mentioned but there were no results, and it was unclear if that was incorporated into the model.

It would be good to have the number of reads that aligned to the background genome.

The methods do not describe exactly how the target amplification efficiency was calculated.

Minor

Line 10 Abstract: Change “Evaluate” to “evaluates”

Page 3 line 5 and 6: Change “ie” to “i.e.,”

Page 3 second paragraph: change “as well as identifying” to “and identified”

Page 3, 4th line of last paragraph in introduction beginning with “First, active learning” this sentence should be reworded. Something like “First, active machine learning was used to identify features that predict primer and primer set efficiency”

Figure 3 caption, last line: change “axis” to “x-axis”

Page 8 Section “Empirical evaluation of primer sets”: Can you clarify that the FspE1 digestion was to suppress mitochondrial amplification and note whether or not swga2.0 has the option to omit mitochondrial sequences?

Page 9 first line: Missing reference and version for samtools.

Page 9, second paragraph, last line: Change “data from the rounds” to “data from rounds”

The text references Figure 3A but there is only a single for Figure 3.

The variables described in Stage 4 of the methods are not described until much later in the paper. I would move them up to where they are first mentioned.

Page 13, line 8: Change “eliminate” to “eliminating”

Page 14: Sentence “Highthroughput sequencing of the amplification products from the three effective P. melaninogenica primer sets reached 1× coverage across 25–64% the target genome with 50Mbp sequencing effort while only 27% of the M. tuberculosis was covered at 1× at similar sequencing effort after amplification with the most effective primer set (Figure 5)” is confusing. Figure 5 does not contain data about M. tuberculosis so I would move the figure reference closer to the actual result and clarify again that you are referring to M. tuberculosis results from the previous version of swga.

Pg 17 line 9: Change “utilizing using” with “using”

Reviewer #2: Yu et al. describe swga2.0, a pipeline that improves upon the original swga program described by Clarke et al. to identify, screen, and choose primers for use in selective whole-genome amplification. The authors apply swga2.0 to identify candidate sWGA primer sets for enrichment of P. melaninogenica and then test the best performing primer sets experimentally.

The manuscript is well-written and easy to follow. Major advances of the new pipeline are greatly reduced computational time (the prior version required weeks on a high-powered computing cluster, the current version purports to requires hours on a personal laptop) and use of machine-learning methods to select primers that are likely to perform well. The model used to predict amplification efficiency (or more accurately, probability of poor amplification) could be very useful for a variety of applications, though experimental data for enzymes other than phi29 would be required. Several companies offer services for highly multiplexed primer design but do not make their code publicly available. The approaches described here could be useful not only for sWGA primer design but for other applications as well, including design of highly multiplexed amplicon panels. The manuscript could be improved in several ways as outlined below – namely application to more than one pathogen and explicit comparison of swga2.0 to the original swga program.

MAJOR

1. Table 2: It is important to emphasize clearly that these features are specific for phi29. Some groups have begun using Equiphi29 to achieve higher reaction temperatures (enabling longer/more specific primers). Others may want to adapt these methods for other enzymes such as Bst. If the features employed by the program are phi29-specific, then the model will probably not perform well for these applications.

2. Page 4: It would be useful to delineate more clearly how swga2.0 differs from the original swga program published by Clarke et al., both in terms of the approach and empirical differences in output. As for approach, in the figure on page 4, there appear to be new steps in boxes 1 (jellyfish), 2 (homodimerization probability, runs of repeats), and most importantly in boxes 3 (random forest model) and 4 (coverage ratio). Some of these differences are covered on page 17 in the Discussion, but a table highlighting differences/improvements would be helpful. As for empirical differences in output, the authors include some discussion of differences on page 18 but are not comparing apples-to-apples. Differences between P. melaninogenica swga2.0 output and M. tuberculosis swga output are interesting, but direct comparison of swga vs swga2.0 for the same organism is needed to delineate their differences clearly. Did the authors run the original sWGA program on P. melaninogenica? If so, how did the output compare? If not, can they run it using similar parameters and comment on differences?

3. How did variant calls in the P. melaninogenica sequencing data compare to reference and to the non-sWGA sample? Phi29 is reported to have a very low error rate, but it is important to confirm with their own data if possible. Another issue of relevance in some cases is the ability to investigate within-host diversity after sWGA – specifically whether allele frequencies can still be estimated with confidence after sWGA enrichment (ex: Oyola et al. PMID 27998271). Though it is probably out of the scope of this manuscript to test experimentally, commenting on these effects in the discussion is probably worthwhile.

4. Confidence in the program’s ability to produce effective sWGA primer sets would be increased if more than one pathogen were tested experimentally. In the original swga publication, Clarke et al. reported differences in sWGA performance across several different organisms. I wonder if the new swga2.0 algorithm might overcome some of these differences, or if they will persist?

MINOR

Page 2: sWGA has now been used successfully for Treponema pallidum subs pallidum (causative agent of syphilis) – see Thurlow CM et al. mSphere 2022, PMID 35491834.

Page 3: Unclear why ‘SWG amplification’ is used instead of ‘sWGA’.

Page 5: Please define each of the ratios included in the score equation listed at the bottom of the page.

Pages 6-7 and Table 2: The included explicit discussion of thermodynamically-principled included in the model is helpful.

Page 8: Why was the sample digested? I assume this is to linearize circular bacterial DNA. Has this been shown to improve the efficiency of MDA/phi29 reactions? In this paper, DNA was digested with FspEI while the previous study using NarI (Clarke et al., 2017). Methylation digestion with different enzymes and experimental process can vary the swga results, such as the proportion of reads mapping to the target organism and genome coverage (see Teyssier NB et al. PMID 33637093).

Page 8: For clarity, please specify exactly what 3.5mM total of sWGA primers means so that there is no ambiguity. Ex: “…(e.g. in a set of 10 primers, each primer was added at 0.35mM so that the total molarity of primers was 3.5mM)” if that is the case, or other wording depending on the actual conditions. There has been variability in published sWGA methods on this topic, due in part to ambiguity in the description of the exact conditions used during sWGA in some past publications.

Page 18: The general suggestions provided are very helpful for scientists looking to use this methodology to design and test their own custom sets.

Page 19: Mention of CRISPR and gRNA design in the concluding sentence doesn’t follow from the manuscript and was surprising. Consider removing, or moving up into the discussion.

Methods: In swga2.0, is it possible to visualize/export exact locations of primer binding sites?

Methods: Is it possible to set the number of mismatched bases permitted in the primers themselves? Can the allowable length of mononucleotide repeats be modified?

**Have the authors made all data and (if applicable) computational code underlying the findings in their manuscript fully available?**

Reviewer #1: **No: **The code is available and well documented but I do not see the sequencing reads, the raw data, or the primer sequences.

Reviewer #2: Yes

PLOS authors have the option to publish the peer review history of their article (what does this mean?). If published, this will include your full peer review and any attached files.

Reviewer #1: No

Reviewer #2: **Yes: **Jonathan B. Parr
---

## [Decision Letter · Decision Letter 1]

9 Feb 2023

Dear Dr. Brisson,

Thank you very much for submitting your manuscript "A fast machine-learning-guided primer design pipeline for selective whole genome amplification" for consideration at PLOS Computational Biology. As with all papers reviewed by the journal, your manuscript was reviewed by members of the editorial board and by several independent reviewers. The reviewers appreciated the attention to an important topic. Based on the reviews, we are likely to accept this manuscript for publication, providing that you modify the manuscript according to the review recommendations.

Sincerely,

Manja Marz

Software Editor

PLOS Computational Biology

Manja Marz

Software Editor

PLOS Computational Biology

Reviewer's Responses to Questions

**Comments to the Authors:**

Reviewer #1: This resubmission is much improved, and the authors have addressed my major concerns. I have a few minor comments.

The authors added Table 1 which helpfully summarizes the improvements in swga2.0 compared to swga1.0 and breaks down the performance improvements for all stages. However, the benchmarking details could be a bit more explicit in a computational biology article. For example, I had to go to the Github page to find swga2.0’s default setting for number of CPUs (which I assume is what was used). After finding out that it defaults to “all” I then needed to look up what that might be on the specific model of MacBook Pro that was used (4 maybe)? I think that this is important information to have in the article to evaluate the 10-fold improvement in speed. Another minor point on Table 1, the first column could include a short name/description of each of the stages in addition to just the stage number so that the reader does not have to refer to the text to understand the table.

My previous comment about the method for calculating amplification efficiency was referring to the empirical values, not the predictions using the model. Some detail about the method used to measured primer amplification of the plasmids would be helpful, even if it is only included in the caption for Supplemental Table 1.

All other issues that I have raised have been addressed by the authors.

Reviewer #2: I appreciate the authors’ responses to my comments and the associated manuscript changes. One minor comment as below:

Discussion: The authors include the following statement: “Also similar to prior results (Oyola et al., 2016), protocols developed with swga2.0 retain the ability to investigate within-host microbial diversity using SWGA enrichment (Pilling et al., 2022), without introducing errors, as the SWGA biochemistry remains identical (Table S4).”

We have found that reliable estimation of allele frequencies after sWGA is not straightforward, due in part to exceptionally high depth of coverage in some regions (“jackpotting”) but not others. If the authors do not have empirical data to support the statement above, I would suggest softening the language to indicate that this is their expectation but not an observation (i.e. replace “retain” and with “are expected” or similar).

**Have the authors made all data and (if applicable) computational code underlying the findings in their manuscript fully available?**

Reviewer #1: Yes

Reviewer #2: Yes

PLOS authors have the option to publish the peer review history of their article (what does this mean?). If published, this will include your full peer review and any attached files.

Reviewer #1: No

Reviewer #2: **Yes: **Jonathan Parr

Figure Files:

Data Requirements:

Reproducibility:

References:

---

## [Editor Report · Decision Letter 2]

23 Mar 2023

Dear Dr. Brisson,

We are pleased to inform you that your manuscript 'A fast machine-learning-guided primer design pipeline for selective whole genome amplification' has been provisionally accepted for publication in PLOS Computational Biology.

Best regards,

Manja Marz

Software Editor

PLOS Computational Biology

Manja Marz

Software Editor

PLOS Computational Biology

---

## [Editor Report · Acceptance letter]

3 Apr 2023

PCOMPBIOL-D-22-00648R2 

A fast machine-learning-guided primer design pipeline for selective whole genome amplification

Dear Dr Brisson,

I am pleased to inform you that your manuscript has been formally accepted for publication in PLOS Computational Biology. Your manuscript is now with our production department and you will be notified of the publication date in due course.

With kind regards,

Timea Kemeri-Szekernyes
